# Monthly Wind Power Forecasting: Integrated Model Based on Grey Model and Machine Learning

**Xiaohui Gao**

School of Management Science and Engineering, Nanjing University of Information Science and Technology, Nanjing 210044, China; 20201124001@nuist.edu.cn

**Abstract:** Wind power generation has been developed rapidly due to rising global interest in renewable clean energy sources. Accurate prediction of the potential amount of such energy is of great significance to energy development. As wind changes greatly by season, time series analysis is considered as a natural approach to characterize the seasonal fluctuation and exponential growth. In this paper, a dual integrated hybrid model is presented by using random forest (RF) to incorporate the extreme gradient boosting (XGB) with empirical mode decomposition (EMD) and a fractional order accumulation seasonal grey model (FSGM). For seasonal fluctuation in vertical dimension processing, the time series is decomposed into high and low frequency components. Then, high and low frequency components are predicted by XGB and extreme learning machine (ELM), respectively. For the exponential growth in horizontal dimension processing, the FSGM is applied in the same month in different years. Consequently, the proposed model can not only be used to capture the exponential growth trend but also investigate the complex high-frequency variation. To validate the model, it is applied to analyze the characteristics of wind power time series for China from 2010 to 2020, and the analysis results from the model are compared with popularly known models; the results illustrate that the proposed model is superior to other models in examining the characteristics of the wind power time series.

**Keywords:** wind power generation; empirical mode decomposition; extreme gradient boosting; grey model; integrated model

## 1. Introduction

The whole world is facing the problem of energy poverty, and consequently, it is urgent to develop clean renewable energy. The expansion of the gap between supply and demand, the depletion of fossil fuel reserves and environmental pollution have forced countries to explore reliable wind power generation [1]. Wind power as a major sustainable energy has made people pay greater attention because of its advantages of safety, renewability, reducing pollution, , and its environmental protection role [2]. The world continues to focus on wind power application, leading the wind power forecasting to play an increasingly central role in the energy scheduling.

Research on wind power has emerged in a big way, and the studies are classified into the following four categories in general: statistical models [3,4], grey model [5], artificial intelligence models [6,7], and hybrid forecasting models [8,9]. Traditional models include multiple linear regression [10], auto regressive moving average (ARMA) [11], and auto regressive integrated moving average (ARIMA) [12]. In order to utilize the characteristics of each model to improve the prediction accuracy, a hybrid prediction method combining wavelet transform and ARMA is proposed by Yang et al. [13]. Erick applied the bagging ARIMA model to predict the electric energy consumption [14]. These single traditional models often lack consideration of unstable patterns existing in wind power time series, and they perform poorly in long-term prediction [15]. Grey model has developed rapidly in recent years. Aiming at the prediction of monthly data, Wang et al. [16] added the seasonal

factors to the GM (1,1) model, which effectively improved the prediction accuracy. In order to reduce the reduction error, Wu et al. [17] proposed the fractional GM (1,1) model and adjusted the change trend of the predicted value by accumulating the order. Actually, the grey model can capture the exponential growth trend in a small sample. As artificial intelligence is further developing, abundant intelligent models are presented, such as support vector machine [18], artificial neural networks (ANN) [19], ELM, RF [20], and long short term memory recurrent neural network (LSTM) [21]. The performance of SVM depends on the selection of kernel function. For a practical problem, the appropriate kernel function needs to be selected according to the actual data model to construct the SVM algorithm [22]. The artificial neural network needs a large number of parameters, and the learning time is too long, even though it may not achieve the purpose of learning [23,24]. To improve forecasting accuracy, Xiao et al. [25] developed a self-adaptive kernel extreme learning machine (KELM). Somu et al. [26] proposed an energy consumption prediction model based on the long short memory network. However, these artificial intelligence models often suffer from a high possibility of being entrapped in local optimization, and having a long computation time and dependence on large sample data [27]. The hybrid forecasting models on wind power time series have gained more attention, including decomposition strategy, artificial intelligence model optimized by grey wolf optimization, etc. Decomposition strategy has become one indispensable step in the wind time series analysis [28]. The common decomposition methods are singular spectrum analysis [29], EMD [30], ensemble empirical mode decomposition [31], and variational mode decomposition [32]. Optimization algorithms include genetic algorithm, particle swarm optimization algorithm, etc. [33,34]. The prediction models include ELM, XGBoost, etc. Furthermore, an ensemble learning model such as XGBoost performs well in complex time series [35]. At present, the hybrid model mainly combines time series decomposition and prediction. Wang et al. [36] proposed VMD and an XGBoost regression model. However, according to the existing literature, many scholars often use a single modeling to predict each IMF when building the prediction model of wind power. A single prediction model cannot fully capture the inherent characteristics of a variety of changes [37,38]. Considering the characteristics of each IMF, the construction of models should be targeted to IMF. On the other hand, data mining is limited to the vertical dimension and the little amount of available data limits the prediction accuracy of single artificial intelligence model because of the short development time of wind power generation [39].Therefore, horizontal dimension data mining should also be paid attention to in the forecasting process.

The wind power forecasting model is generally established through data mining technology based on historical operation data. From the point of current research, wind power generation forecasting is a small sample and a complex trend prediction problem in essence [40]. Scholars often focus on the vertical processing while ignoring the horizontal processing of the data. The horizontal dimension data are obtained by sampling at the interval of the cycle. In fact, data processing from a comprehensive perspective has a crucial guiding significance for the model selection. Moreover, mining the change characteristics of the sample data in both the vertical horizontal dimension can also effectively improve the prediction accuracy of the model. Decomposition strategy becomes one of the important steps in the wind power time series. This paper proposes an integrated hybrid model which fully considers the data change characteristics of horizontal and vertical dimensions. In the primary stage, EMD-XGB-LEM and FSGM models are used to process the initial sample data vertically and horizontally. In the secondary stage, RF is used to integrate the results of the two models in the first stage. The contribution of this paper can be summarized as the following two points:

- From horizontal dimension, the FSGM is superior to a seasonal grey model in adjusting the growth trend by changing the parameter;
- From the vertical dimension, the wind power time series fluctuation information extracted by the EMD-XGB model;

- The advantages of the two models can be fully integrated, digging and utilizing more information comprehensively.

The remaining sections of this paper are organized as follows: In Section 2, the architecture of the wind power generation forecasting model is introduced. The application of hybrid model is then empirically studied in Section 3. Finally, Section 4 concludes the study.

## 2. The Methodology

### 2.1. Background

The wind power generation has developed rapidly, containing seasonal fluctuation and exponential growth in the wind power time series [41]. The aim of the integrated prediction model proposed in this paper is to improve the prediction accuracy by combining the prediction models from vertical and horizontal dimensions. For example, the sequence of point $A$ in year $T$ is not only related to point $A - 1$ in year $T$, but may also be affected by point $A$ in year $T - 1$. Then, there are two modeling ideas: One is based on the continuous evolution of the time series, such as the time series of $A, A + 1, A + n$ in T to predict $A + n + s$ in T. The other is to model based on the year-on-year evolution of the time series, such as the time series of point $A$ in $T - n$ year, point $A$ in $T - n + 1$ year, and point $A$ in $T + s$ year. This paper defines these two modeling ideas as vertical processing and horizontal processing, respectively, by using EMD-XGB-ELM and FSGM.

EMD is an adaptive signal decomposition algorithm for complex signals proposed by Huang et al. [42]. After decomposing the noisy signal, the noise and the effective signal are separated into different IMFs. The signal is reconstructed by reasonable utilization of IMF to achieve the purpose of noise removal [43]. Extreme gradient boosting, the integration of many classification and regression trees, is a scalable tree lifting algorithm system [44]. At present, XGB has been widely used in competition, finance, and other fields. In this algorithm, the regular term is introduced into the loss function to prevent over fitting of the model, processing a large amount of data more quickly and effectively. The GM (1,1) model has been widely used in our daily production and life, solving the problem of modeling small sample data [45]. A seasonal grey model is suitable for data with an exponential growth trend. The fractional order accumulation can not only smooth the irregularity of data but also reduce the disturbance of the grey model solution and improve the prediction accuracy. Therefore, FSGM is used to deal with the wind power series prediction with an exponential growth trend. The Lempel–Ziv complexity algorithm is proposed to represent the rate of new patterns appearing in time series. Considering that the signal needs to be fine-grained before calculating the complexity, this paper adopts binary coarse-grained. For more details of each method, see the reference [46].

### 2.2. Modeling Strategy

The wind power time series is affected by many factors, and there are random interference signals, long-term trend components and seasonal changes due to temperature, wind speed and other factors. It is difficult to fit the trend of change by a single prediction model. The proposed hybrid model integrates the advantages of grey model and intelligent algorithms. Taking the error as the objective function, the accuracy and stability of the model are considered. The overall framework of the proposed hybrid model is shown in Figure 1. To be more specific, in the first stage: EMD is used to decompose the original data in the vertical dimension, and then all IMFs are divided into high frequency and low frequency. XGB is used for high frequency, and ELM is used for low frequency prediction. The horizontal dimension data are obtained by sampling at the interval of cycle $T$, and the forecast data are obtained by modeling with the FSGM model. In the second stage, take the two results obtained in the first stage as new inputs and integrate them with RF to obtain the final prediction results.

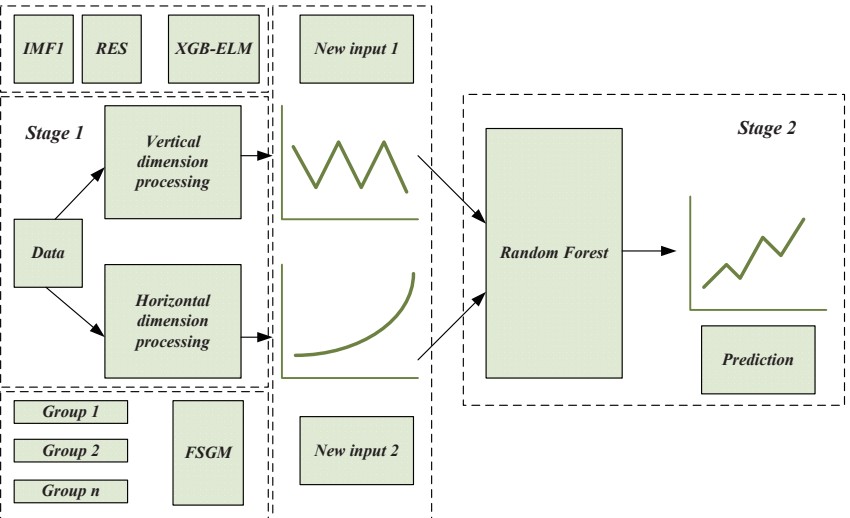

**Figure 1.** The framework of the proposed model.

This paper proposes a new modeling strategy. According to the vertical and horizontal changes of time dimension, the original data are divided into two sample processing groups. Among them, the sample data of the vertical processing group are arranged by the monthly increasing order, and the sample data of the horizontal processing group are arranged by the monthly increasing order of the same period. Then, the EMD-XGB-ELM model and FSGM model were constructed, and the RF was used to integrate two models. In this way, the output data not only retain the vertical variation characteristics of the original data, but also retain its horizontal variation characteristics.

In view of the highly irregular and volatile characteristics of longitudinal processing group sample data, the Lempel–Ziv complexity algorithm is introduced into the EMD-XGB-ELM model on the basis of high and low frequency classification of IMF and residuals after decomposition. Average mutual information (AMI) is further introduced to obtain a data input matrix by utilizing the time lag effect of the wind power time series. In addition, this paper introduces the fractional order accumulation operator into a traditional SGM model to build an FSGM model, aiming at the poor information characteristics of short sample data in the horizontal processing group.

*2.3. Modeling Process*

In this section, the prediction results of EMD-XGB and FSGM are combined by random forest. The detailed framework of the proposed hybrid model is displayed in Figure 2. Figure 2 shows that the prediction process is divided into two stages. In the first stage, EMD-XGB- ELM and FSGM models are respectively used for modeling to obtain prediction data. In the second stage, use the results of the first stage as RF input to obtain the final predicted value. The entire forecasting process is described as follows:

Step 1: Obtain the initial data set, using 90% of the data set as the training set and 10% of the data set as the test set.

Step 2: The data in vertical dimension are $X_T(t)_Z = \{x_{2010}(1), \cdots, x_{2020}(12)\}$, and the data in horizontal dimension are $X_T(t)_H = \{x_{2014}(1), \cdots, x_{2020}(1)\}$.

Step 3: EMD is used to decompose the training set of the vertical dimension, and the Lempel–Ziv complexity algorithm is used to identify the high frequency component (HF) and low frequency component (LF) of the IMF. For the training set of the horizontal processing group, the seasonal factor of the training set of the horizontal vertical dimension is calculated, and the fractional order accumulation is calculated to obtain the $r$-order $(0 < r < 1)$ accumulation generating sequence $X_s^{(r)} = \{x_{s1}^{(r)}, x_{s2}^{(r)}, \cdots, x_{sn}^{(r)}\}$.

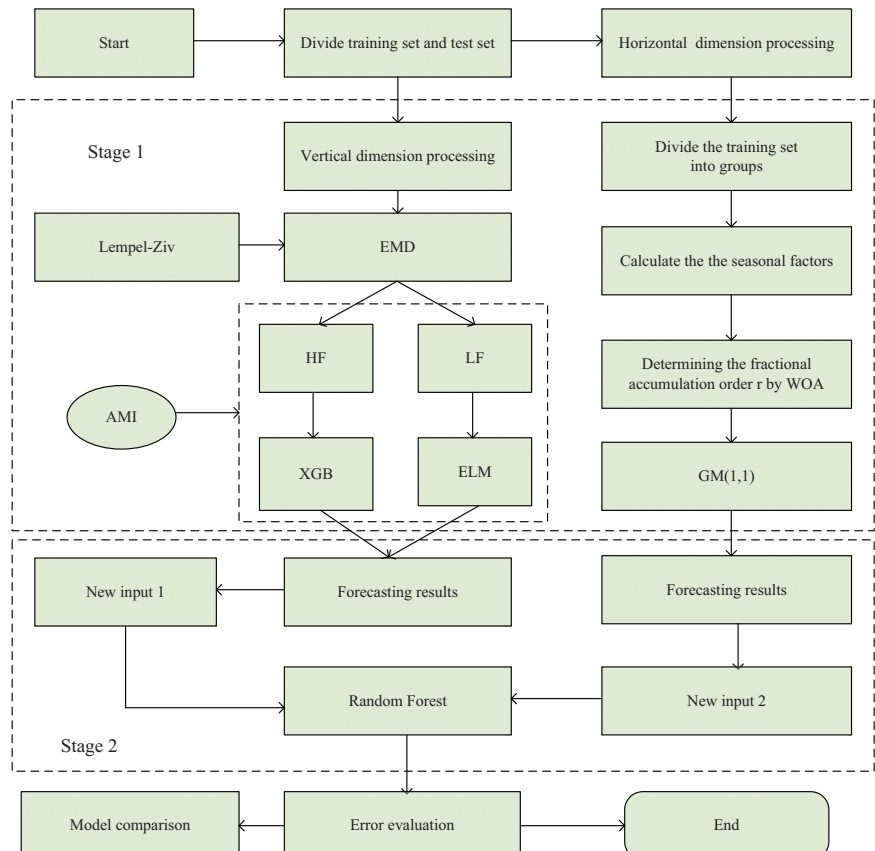

**Figure 2.** Wind power forecasting modeling based on RF.

Step 4: AMI is used to calculate the time delay of IMF and residual in the the vertical dimension. The historical time series are used as the feature inputs of the model according to the time delay $\{\tau_1, \tau_2, \cdots, \tau_s\}, \{\tau_{s+1}, \tau_{s+2}, \cdots, \tau_m\}$. The GM (1,1) model is constructed by $X_s^{(r)}$ and the least square method is used to estimate the parameters in GM(1,1) model. The whitening differential equation of GM (1,1) is solved. Then, the time response formula is obtained to obtain the r-order cumulative predictive value $\hat{X}_s^{(r)}$. The $\hat{X}_s^{(r)}$ is multiplied by the corresponding seasonal factor, and the cumulative factor is restored to obtain the final predicted value of $\hat{X}$.

Step 5: The prediction result $\{HFs_{p1}, HFs_{p2}, \cdots, HFs_{ps}\}$ is calculated by using XGB. The LF prediction result of $\{HFs_{ps+1}, HFs_{ps+2}, \cdots, HFs_{pm}\}$ can be obtained by using the ELM. Then, integrate all the component results to obtain $\hat{x}_{m+1}$. Update the data with the next phase of the test set to obtain the training set $\{x_2, x_3, \cdots, x_{m+1}\}$. The prediction value can be obtained $\hat{x}_{m+2}$ by repeating the vertical processing operation from step 3 to step 5 until the prediction of all test sets in the processing is completed.

Step 6: The prediction results obtained by EMD-XGB-ELM and FSGM in the primary processing stage are used as the input data of RF to obtain the final prediction results.

Generally speaking, there are three error evaluation indicators, namely, root mean square error (*RMSE*), mean absolute error (*MAE*), and mean absolute percent error *(MAPE)*. These three error evaluation indicators are used to verify the performance of model comprehensively. The calculation formulas are described as follows:

$$MAPE = 100\% * \frac{1}{n} \sum_{i=1}^{n} \left| \frac{x^{(0)}(k) - \hat{x}^{(0)}(k)}{x^{(0)}(k)} \right| \tag{1}$$

$$RSME = \sqrt{\frac{1}{n}\sum_{i=1}^{n}(x^{(0)}(k) - \hat{x}^{(0)}(k))^2} \qquad (2)$$

$$MAE = \frac{1}{n}\sum_{i=1}^{n}\left|x^{(0)}(k) - \hat{x}^{(0)}(k)\right| \qquad (3)$$

where $x^{(0)}(k)$ is the actual value, and $\hat{x}^{(0)}(k)$ is the predicted value that indicates the actual value and the forecasted value, respectively.

## 3. Case Study

In this paper, the research object of China's wind power generation is accessed on 25 August 2022 cited in https://data.stats.gov.cn/ (Unit:Billion Kwh). All experiments were carried out on Python (3.8) and MATLAB (2019a) on Windows 10.

### 3.1. Forecasting the Wind Power Generation in China

3.1.1. Vertical Dimension Processing

The original data are vertical dimension data, and the data sample by period 12 is horizontal dimension data. The Hurst index of China wind power time series is shown in Figure 3. The slope of the straight line in Figure 3 represents the value of Hurst index. It can be seen that the Hurst value is 0.99. The value is close to 1, indicating that the series has long memory.

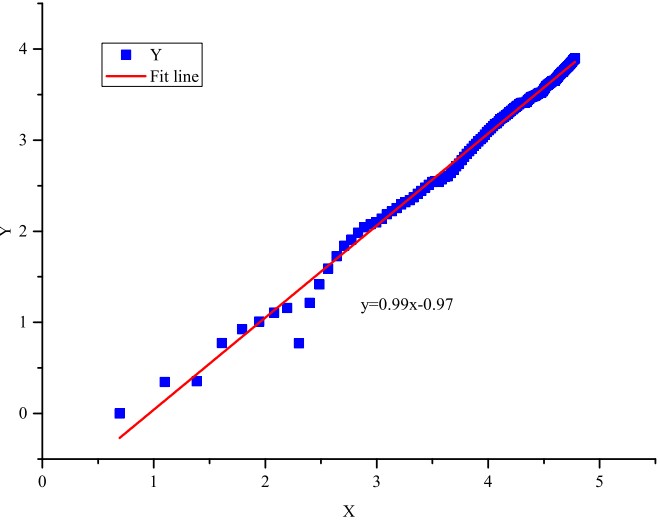

**Figure 3.** The rescaled range analysis of the wind power time series.

First, EMD is used to decompose the training set, and the decomposed components are arranged according to the complexity from high to low. According to Figure 4, the time series of wind power in the vertical processing group can be decomposed into three IMFs and one residual term (RES).

The different data characteristics of high and low frequency components represent the different internal characteristics of time series. For each IMF (including high and low frequency components), the high frequency component has the greatest impact on the final integrated prediction results. This is because the change trend of high frequency component is the closest to the original sequence, and it is also the most difficult to deal with. In contrast, the low frequency component is considered as a stationary sequence because of its small influence. Based on this, this paper uses the Lempel–Ziv complexity algorithm to identify the high and low frequency of the decomposed components of EMD, and then matches the optimal prediction model of each component. From Figure 5, it can be found that IMF1, IMF2, and IMF3 are high-frequency components, and the RES is the low-frequency component. According to the fluctuation characteristics of high and low

frequency components, XGB is used to predict high frequency components, and ELM is used to predict low frequency components.

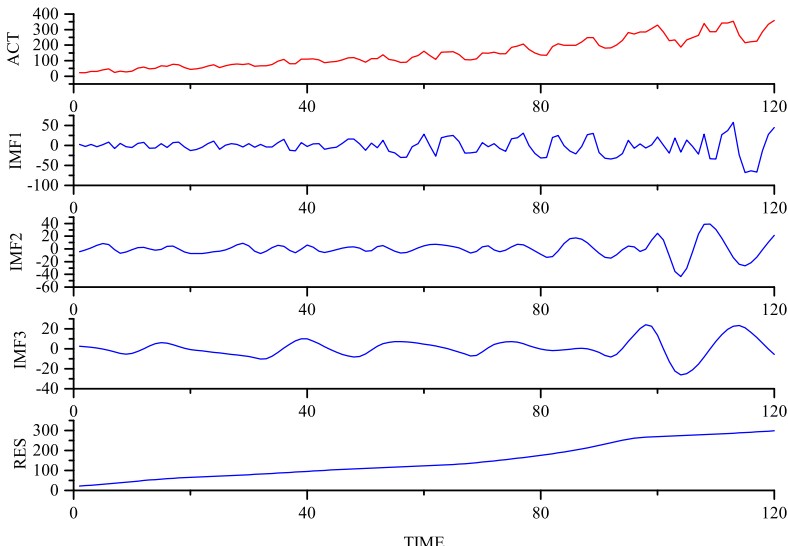

**Figure 4.** The EMD decomposition results of wind power time series.

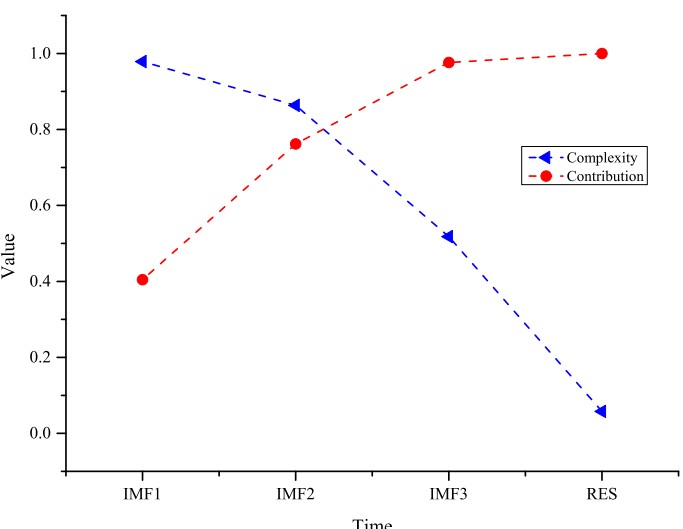

**Figure 5.** The complexity of each component and its accumulation contribution.

From the perspective of information theory, AMI is suitable for the analysis of non-linear systems. Its basic idea is to select the time when the mutual information function reaches the local minimum for the first time as the best IMF delay time. According to the time delay of each IMF, each matrix form is obtained as the input data of the prediction model. Considering the influence of time delay effect on long sample time series prediction, this paper further uses AMI to calculate the delay time. Among them, the high-frequency component takes IMF1 as an example, and its time lag effect is shown in Figure 6 . It can be seen from Figure 6 that the time delay of IMF1 is 5, that is, the observed value of IMF1 in the first five days is used to predict the value of IMF1 on the sixth day, and the matrix form obtained from this is used as the data input of the prediction model. By analogy, the time delay of all high and low frequency components is obtained; then, the data input of each component is obtained.

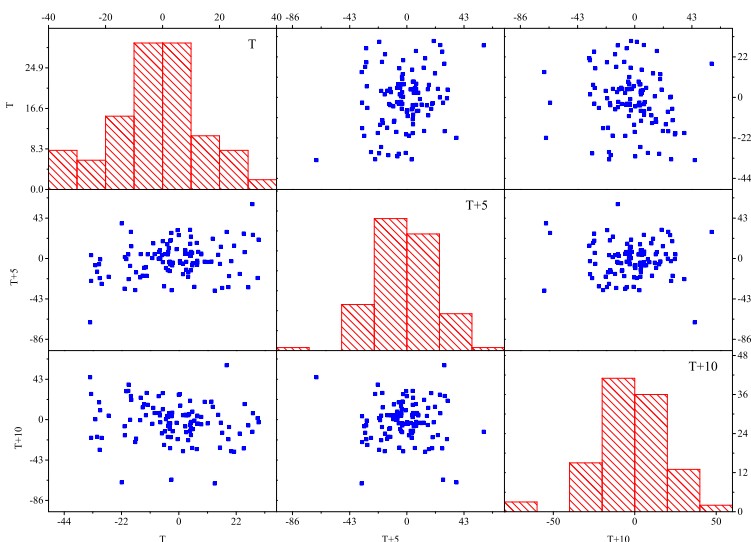

**Figure 6.** The time delay diagram of IMF1.

### 3.1.2. Horizontal Dimension Processing

From the horizontal dimension, the wind power generation is shown in Figures 7 and 8. From Figures 7 and 8, we can see that the data show a strong regularity, that is, every month shows a growth trend from 2010 to 2019. The wind power series has an upward trend, but the magnitude of the increase needs to be determined according to the FSGM. Compared with the traditional SGM model, FSGM is more accurate in describing the range of data changes. Through order r to provide the change trend of different amplitude, we can choose the change amplitude which is more suitable for the actual situation. In order to determine the optimal parameters of r, the whale optimization algorithm is used to optimize the selection.

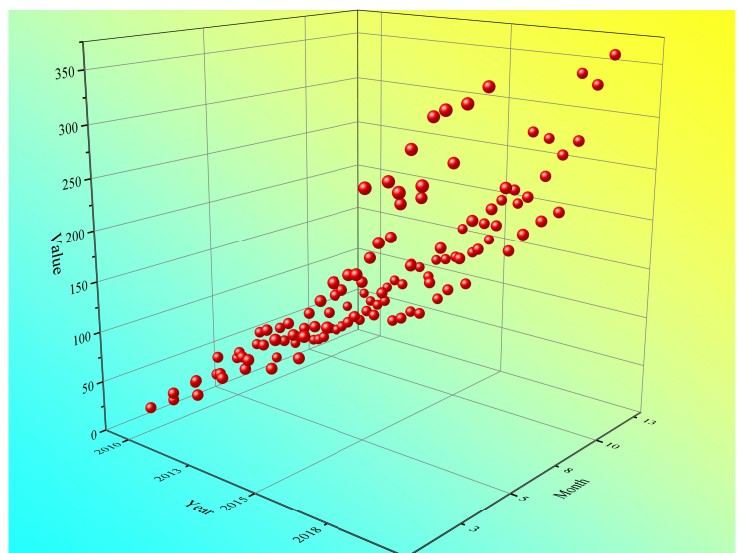

**Figure 7.** The three-dimensional diagram of monthly wind power time series.

FSGM adjusts the change trend by accumulating order *r* to obtain a more accurate change trend and describe the internal change characteristics of data more accurately.

Figure 9 is the relationship between the iteration and r. From Figure 9, we can see that, when the iteration is 200, the value of r is 0.56—that is to say, when *r* = 0.56 is more in line with the actual situation.

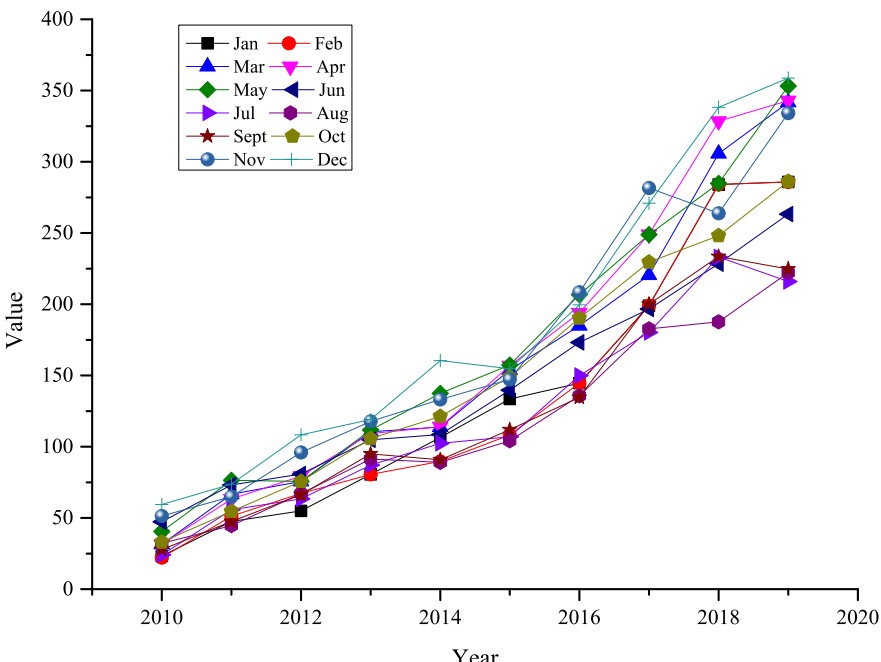

**Figure 8.** The horizontal dimension of monthly wind power time series.

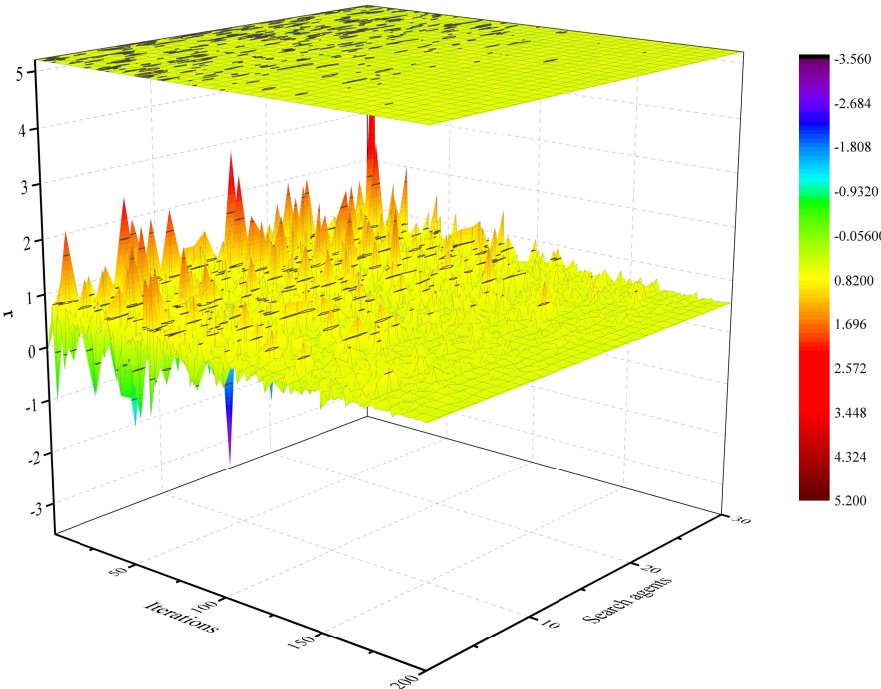

**Figure 9.** The optimization process of WOA.

### 3.1.3. Forecasting Results with RF

Figure 10 depicts the results of hybrid model in the fitting period. The more points in the graph are concentrated on the line $y = x$, the better the fitting performance is. It can be seen from Figure 10 that these points are concentrated around this line, and the performance of the hybrid model is better.

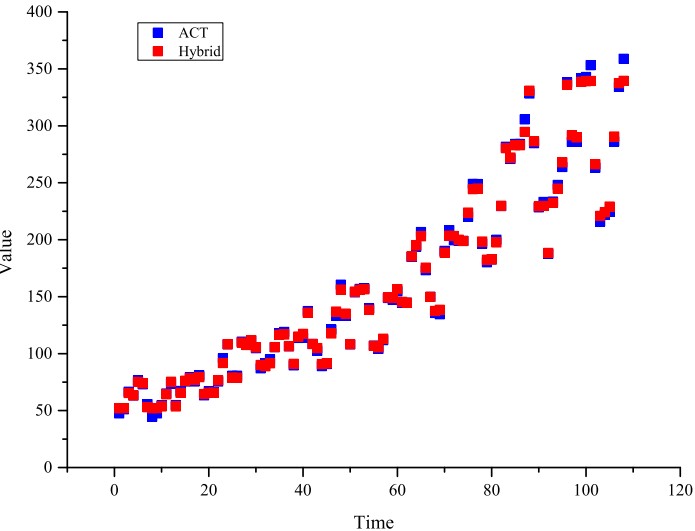

**Figure 10.** The results of the hybrid model in the fit period.

### 3.2. Comparison of Results with Other Models

To comprehensively evaluate the performance of the hybrid model, it is compared with several forecasting models, including HW, SARIMA, SVM, LSTM, FSGM, EMD-LSTM, EMD-XGB, and RF. In the comparison, the model parameters are the default parameters.

The prediction results and real values of each benchmark model in the prediction period are shown in Figure 11. It can be seen that the accuracy of combination model is higher than that of other models. Figure 12 shows the error of each model in the forecast period. The proposed hybrid model considerably outperforms other models.

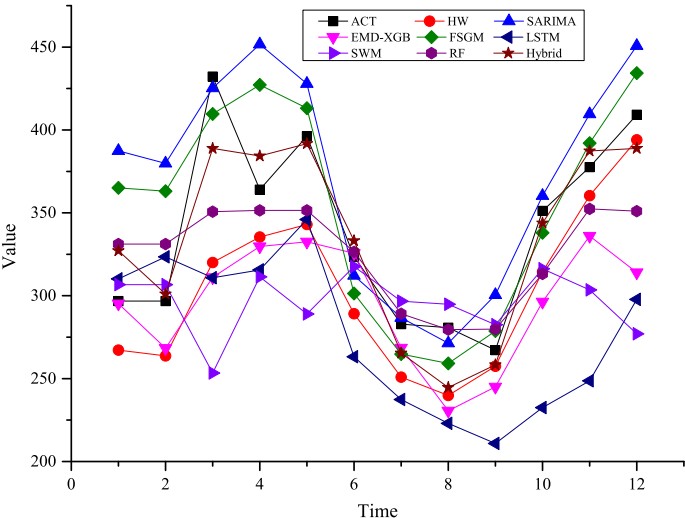

**Figure 11.** The comparison of models in the forecasting period in 2020.

Table 1 reports that the prediction of each model is an error evaluation index. It can be found that LSTM, RF, EMD-XGB-ELM, and the combination model are better models in the fitting period, and RF, EMD-XGB-ELM and combination model are better models in the prediction period. However, LSTM, RF, and EMD-XGB-ELM performed better in the fitting period, but the accuracy of the combined model was inferior to that of the combined model in the prediction period. The table indicates that the minimum MAE, RMSE, and MAPE values of the proposed model are 0.64, 0.51, and 0.35 in the fitting period, and 21.57, 17.67, and 5.28 in the forecasting period, respectively. Its prediction accuracy and stability are significantly better than those of the seven other comparison models. The RMSE, MAE, and MAPE values of the combined model were significantly lower than those of other models

in both the fitting and forecasting period. The advantage of the proposed model is that it can comprehensively mine data information to improve the prediction accuracy.

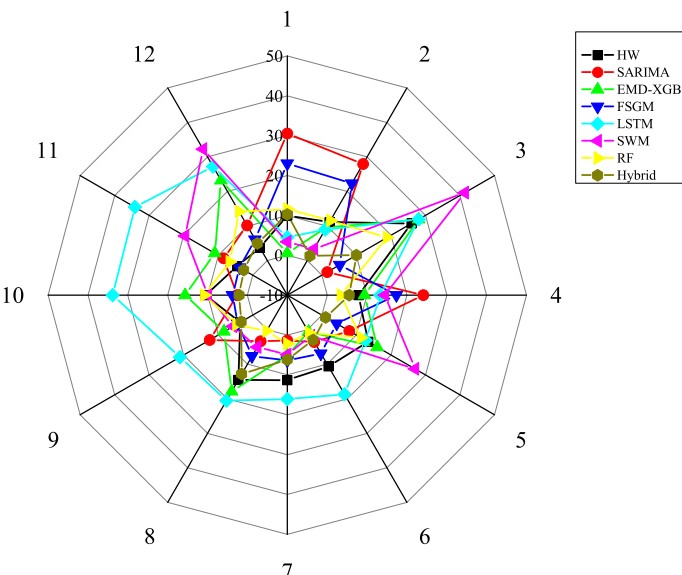

**Figure 12.** The MAPE of each model in the forecasting period.

**Table 1.** The error of each model in the fit and forecasting period.

| Model | Fit | | | Forecasting | | |
|---|---|---|---|---|---|---|
| | *RMSE* | *MAE* | *MAPE* | *RMSE* | *MAE* | *MAPE* |
| LSTM | 0.96 | 0.81 | 0.72 | 79.48 | 69.91 | 19.92 |
| SVM | 2.83 | 1.67 | 0.98 | 77.07 | 54.07 | 14.08 |
| EMD-LSTM | 0.94 | 0.80 | 0.68 | 65.42 | 53.89 | 13.26 |
| EMD-XGB | 0.85 | 0.69 | 0.59 | 56.07 | 44.12 | 12.12 |
| SARIMA | 17.04 | 12.63 | 7.98 | 48.32 | 36.71 | 11.17 |
| HW | 18.45 | 12.93 | 8.27 | 44.76 | 36.90 | 10.60 |
| FSGM | 13.35 | 9.86 | 6.82 | 36.83 | 30.28 | 9.26 |
| RF | 1.12 | 0.74 | 0.65 | 37.46 | 29.32 | 8.06 |
| Hybrid | 0.64 | 0.51 | 0.35 | 21.57 | 17.67 | 5.28 |

## 4. Conclusions

The high complexity and randomness of wind power time series increase the difficulty of accurate prediction, which shows that it is necessary to find a reliable and effective prediction method. In view of the growth trend and seasonal fluctuation characteristics of a monthly wind power time series, this paper proposes a new integrated forecasting modeling idea based on the paradigm of an artificial intelligence model combined with a statistical model suitable for short sample forecasting from the perspective of dual processing of vertical and horizontal dimensions. According to this idea, EMD-XGB-ELM and FSGM are combined by RF, and the monthly wind power time series in China is predicted. In order to verify the effectiveness of the proposed model, it is further compared with HW, SVM, LSTM, FSGM, SARIMA, RF, and EMD-XGB-ELM models. Through the comparative analysis of the models, it is found that the prediction accuracy of the new integrated model based on dual processing perspective is much higher than other benchmark models, which indicates that this model can effectively predict the wind power time series. Moreover, this new integrated forecasting model based on the paradigm of artificial intelligence model combined with a statistical model for short sample forecasting also has good application prospects in other fields, such as stock price, futures price, power load, and so on, which will be the future development direction of this research.

**Funding:** This work was supported in part by the Postgraduate Research and Practice Innovation Program of Jiangsu Province (KYCX21-1034).

**Institutional Review Board Statement:** Not applicable.

**Informed Consent Statement:** Not applicable.

**Data Availability Statement:** Not applicable.

**Acknowledgments:** We would like to thank the editor and reviewers for their valuable comments on this research.

**Conflicts of Interest:** The author declares no conflict of interest.

## Abbreviations

| | |
|---|---|
| ARIMA | auto regressive integrated moving average |
| ARMA | auto regressive moving average |
| ANN | artificial neural networks |
| ELM | extreme learning machine |
| EMD | empirical mode decomposition |
| FSGM | fractional order accumulation seasonal grey model |
| KELM | kernel extreme learning machine |
| LSTM | long short term memory recurrent neural network |
| RF | random forest |
| SGM | seasonal grey model |
| XGB | extreme gradient boosting |

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
