# Peer review of "Monthly Wind Power Forecasting: Integrated Model Based on Grey Model and Machine Learning"

_sustainability, doi:10.3390/su142215403_

Round 1

Reviewer 1 Report

The article describes the problem of predicting energy generation from wind turbines based on an integrated model (combination of gray model and machine learning). This issue is a very current problem and the importance of this type of analysis will certainly increase due to the growing demand for energy from renewable sources, which entails the installation of many large wind farms. The share of wind energy in the overall energy mix must remain under control in order not to cause problems with energy security. For this reason, the topic discussed in the article is important and very topical.

In the introduction, the authors conducted an in-depth analysis of the issue, indicated the current state of knowledge and the disadvantages of the currently used methods. Then, they presented the concept of a new approach to solving the problem in a very detailed, but at the same time legible manner. It should be emphasized that all of this has been clearly and logically described.

Finally, a comparative analysis of various methods, including the proposed one, is presented. The obtained result confirmed the correctness of the theses.

The work was written really well and with great care. I did not find any substantive or editorial flaws. The graphics used are very legible and of good quality. The cited literature does not raise any objections either.

Congratulations to the authors of a great article!

Reviewer 2 Report

The comments are as follows: 

1. There are a few typo errors like on page 2, line 53. 

2. The list of abbreviation is mandatory. 

3. How your work is different from the following work? 

A new hybrid model for wind speed forecasting combining long short-term memory neural network, decomposition methods and grey wolf optimizer Aytaç Altan, Seçkin Karasu, Enrico Zio; Applied Soft Computing Volume 100, March 2021, 106996 

Reviewer 3 Report

This paper has an interesting topic, but there are some points that should be observed

1- More references from microgrid and renewable energy to be discussed. You can use the following references

https://doi.org/10.3390/su141811731

https://doi.org/10.3390/su14106183

https://doi.org/10.1109/JSYST.2021.3077213

2-Regarding figure 1, it is better to provide more explanations about stage1 and stage2

3-Provide more explanations about Figure 2, Figure 7 and Figure 8

4-Compare your proposed method with other similar methods and check its advantages and disadvantages

Reviewer 4 Report

The paper looks more like an internal memo report rather than a paper, and it is quite complicated to follow. It is not clear why the authors are conducting the study and what they are trying to achieve. The introduction and methodology sections should be rewritten and blend together with the results section in a better way. The conclusions are too hastily put together, and there is no evidence that the framework performs well compared to the real world. I suggest the authors to revisit their paper if they want to consider it for publication.

The English language throughout the paper is sometimes difficult to read. Articles are not used properly. I suggest revising the English language of the work.

·         The first paragraph is way too general and add little value to the work: I suggest removing lines 20-33 and rephrase line 34 to give the general motivation of the work in one single sentence.

·         Lines 35-38 : until here it is very difficult to understand where the focus of the paper is: I suggest rephrasing it and get to the point quickly.

·         Lines 85-86: Horizontal and vertical processing seems to be key in the paper, and it could be explained a bit more in detail (one or two sentences, no more).

·         The introduction is structured in a correct way, but it goes very much in detail which might be confusing to readers. I suggest giving the introduction a re-write and try to structure it in this way:

o   Motivation of the work: wind energy importance and development (one single sentence);

o   Background to this study: defining the field of study addressed in the introduction;

o   Literature review: to identify gap in the research addressed by authors, NOT give a summary of the literature. Rather than explaining all the different methods limitations, it would be more appropriate to focus on the family of methods similar to what used in the paper and state what problems are there.

o   Research gap (it is quite unclear what is the gap the authors want to work on, too many problems are mentioned in the literature review).

o   Scope of work (try to summarise lines 97 to 105 in a single catchy sentence): define the novelty of the work in trying to solve the gap from the literature.

o   Structure of paper.

·         Section 2 is a summary of a textbook, for the methodology of the work, more detail into what the authors have applied is needed. Have the authors compared the three referred methods? Why have they chosen these methods? What they are planning to find using the methods (this is also unclear from the introduction).

·         Section 3 is more similar yto a methodology. Probably it is worth merging Sections 2 and 3 into one single methodology and subtitle section 2.1, 2.2 and 2.3 into a background to the study, as it report on not original methods from the literature (I understand the framework is the original bit? This should be made clearer).

·         Section 4.1: why is this section in the results section? It still looks like a textbook more than a paper.

·         Lines 228-229: this bit seems interesting, but why are the authors reporting it? Was it not known before (I doubt it wasn’t). Is it a way of reporting results later on (in that case this part should be moved into the methodology).

·         Section 4.2.1: What is the vertical dimension? Is it referring to the change in height of the wind speed? Perhaps the vertical wind speed component: not clear what the authors are talking about in this paragraph.

·         Figure 5 and following: what is ‘Value’ on the ordinate axis?

·         It is difficult to follow the results section without a robust methodology section explaining the work undertaken.

·         Very few references from MDPI, so it is questionable whether this might be the right journal for this paper.

Round 2

Reviewer 3 Report

Dear Authors

Thanks for addressing my previous comments

Best Wishes

Reviewer 4 Report

Thanks to the authors for addressing all of my comments.

I still think this is quite a weak piece of work, but I am seeing that other reviewers think this is worth publication.

I have no further suggestions for improvement at this stage.